# Strain Measurement during Quasi-Static and Cyclic Loads in AL-6XN Material Using Digital Image Correlation Technique

**DOI:** 10.3390/ma17153697

**Published:** 2024-07-26

**Authors:** Donovan Ramírez-Acevedo, Ricardo Rafael Ambriz, Christian Jesús García, Cesar Mendoza, David Jaramillo

**Affiliations:** Instituto Politécnico Nacional CIITEC-IPN, Cerrada de Cecati S/N Col. Sta. Catarina, Azcapotzalco 02250, Mexico; donovanramirez10@gmail.com (D.R.-A.); cjgarcia@ipn.mx (C.J.G.); cmendozago@ipn.mx (C.M.); djvigu@gmail.com (D.J.)

**Keywords:** strain monitoring, AL-6XN, loading–unloading, low cycle fatigue, digital image correlation

## Abstract

A customized digital image correlation (DIC) system was implemented to monitor the strain produced in a cold-rolled AL-6XN stainless steel plate, 3.0 mm thick, subjected to quasi-static and cyclic loading tests. A comparison of the DIC strain measurements was made against those provided by conventional extensometers. Furthermore, the DIC system was used to monitor the fatigue crack initiation in low-cycle fatigue tests. The true stress–strain behavior for the AL-6XN material was properly captured by the DIC measurements. For low-cycle fatigue tests (strain control), the strain mapping generated by DIC allowed for identifying zones with higher strain than the nominal strain amplitude applied ($\varepsilon_{a}$) since the first stages of the fatigue life (FL). These zones become potential fatigue crack initiation sites.

## 1. Introduction

AL-6XN alloy is a relative new material which was developed by Allegheny Technologies Incorporated (ATI) as a super-austenitic, nitrogen-bearing stainless steel, having an excellent formability and without a plasticity-induced martensitic transformation under cold working. It was designed with a significantly higher content of chromium, nickel, and molybdenum for an improved strength to highly corrosive environments, in comparison to 304 L, 316 L and 317 L stainless steel grades [1]. The American Society for Testing and Materials (ASTM) has included it in the B688 standard specification for chromium–nickel–molybdenum–iron products with the UNS N08367 designation [2]. The AL-6XN alloy is used in a variety of applications, for instance, in the chemical industry and water piping in nuclear plants, where it presents an excellent resistance to chloride, crevice, pitting corrosion, and stress corrosion cracking in both acidic and alkaline environments [3].

Several studies on AL-6XN have been conducted to determine its fatigue behavior at room temperature [4,5], as well as its thermomechanical response in compression tests [6] and low-cycle fatigue (LCF) under different strain rates and temperatures [7,8], including dynamic strain aging (DSA) regime (573 and 873 K). Other studies have been focused on subjects such as the microstructural evolution during the hot-rolling process [9], the effect of dislocations and persistent Lüders bands (PLBs) in the fatigue damage [10], as well as the fatigue behavior in similar and dissimilar welds [11,12,13].

The understanding until now about the fatigue process can be distinguished into three stages: a threshold one, where the cyclic loading accumulates a microscopic internal damage (I); followed by a crack initiation and its stable propagation stage (II); and the final fracture, with an unstable crack growth process, where the crack length reaches a critical size $a_{c}$ (III) [14].

In recent years, Structural Health Monitoring (SHM) has been developing to detect the fatigue damage timely, using optical fibers or a network of ultrasonic transducers for the strain measurements [15]. Due to the physical phenomenon involved in the stage I, which occurs inside of the materials at a microscopic level, the evaluation at the threshold stage of the fatigue damage has remained as one of the biggest challenges. Some researchers have attempted to monitor the stage I by non-destructive techniques, based on quantitative changes in the physical and mechanical properties of the material throughout its fatigue life. Fredrik Bjørheim et al. [16] made a summary of several proposed techniques to detect the accumulated fatigue damage prior to the macroscopic crack initiation, as well as during the fatigue crack growth. Table 1 shows a summary of these techniques.

On the other hand, in the last few decades, the digital image correlation (DIC) technique has taken quite a bit of interest in the scientific community since it provides non-contact full-field deformation data. This technique was developed in the 1980s at the Department of Mechanical Engineering in the University of South Carolina [17,18,19]. It considers a given number of subsets from a region of interest (ROI), where one point P is established in each subset before the deformation $f\left(P\right)=f\left(x,y\right)$ and then tracked to the new position P* after deformation $f*\left(P*\right)=f*\left[{x+u\;(P)}_{,\;\;}\;y+v\;\left(P\right)\right]\;$(Figure 1). Then, both subsets are compared by a cross-correlation coefficient C [17]:(1)$$C\;(u,\;v,\;\frac{\partial u}{\partial x},\;\frac{\partial u}{\partial y},\;\frac{\partial v}{\partial x},\;\frac{\partial v}{\partial y})=\frac{\int_{∆M*}f\left(x,\;y\right)f*\left(x+\xi ,\;y+\eta \right)dA}{{\left[\int_{∆M}{\left[f\left(x,\;y\right)\right]}^{2}dA\int_{∆M*}{\left[f*\left(x+\xi ,\;y+\eta \right)\right]}^{2}\;dA\right]}^{1/2}}$$
where:

ΔM = Subset in undeformed image

ΔM* = Subset in deformed image



$\xi =u+\frac{\partial u}{\partial x}∆x+\frac{\partial u}{\partial y}∆y$





$\eta =v+\frac{\partial v}{\partial x}∆x+\frac{\partial v}{\partial y}∆y$



Due to its versatility, DIC has become a popular measurement technique for a variety of experiments [20,21,22,23,24,25,26,27,28]. However, the use of DIC techniques had been mostly limited to quasi-static conditions. A few manuscripts reported the DIC technique applied to monitor fatigue crack growth processes [26,28]. For instance, Valanduit et al. [26] used the DIC technique in combination with a stroboscopic illumination source to monitor the fatigue crack growth process. They could analyze fatigue tests conducted at several frequencies and up to 12 Hz. On the other hand, Niendorf et al. [22] and Risbet et al. [25] carried out studies to monitor fatigue damage (stage I) during low-cycle fatigue tests; in all these works, no validation of DIC strain measurements against conventional techniques, like extensometers, were performed.

The objective of the present work was to use the DIC technique to monitor the generated strain in two dimensions (2D) during quasi-static and cyclic loading tests of AL-6XN specimens. A comparison was made for the strain measured by a conventional physical extensometer and a virtual extensometer set by the DIC technique, which used a low-cost and customized hardware system comprising a conventional cell phone armed with a 64 megapixels camera and a free basic software for the image analysis. In addition, the DIC-customized system was used for full-field strain measurements during a fatigue test in order to detect the fatigue crack formation.

## 2. Materials and Methods

A super-austenitic stainless steel plate (AL-6XN) in the annealed condition was used (700 mm long by 300 mm width and 3 mm thick). A chemical analysis was conducted as follows: Mn, P, Si, Cr, Ni, Cu, Mo, and Fe by optical emission spectroscopy (OES); C and S by combustion technique; and N by thermal conductivity (Table 2).

To analyze the microstructure, metallographic analyses were performed in the longitudinal (L), short transverse (ST), and long transverse (LT) surface directions regarding the rolling direction. Metallographic specimens (10 × 10 × 3 mm) were prepared following the guidelines described in ASTM E3 [29]. Polished specimens were etched using glyceregia as a chemical reagent, according to ASTM E407 [30], and then observed with a MA200-eclipse Nikon optical microscope.

Rockwell B hardness tests were carried out by making ten random indentations on L, ST, and LT surface directions. The indentations were performed on a digital Wilson Rockwell hardness tester model 574T in accordance with ASTM E18 [31] standards. 

To evaluate the DIC-customized system and obtain the stress–strain behavior, tensile tests were conducted according to ASTM E8 [32]. Sub-size specimens were obtained on the LT direction with a gauge length of 25 mm (Figure 2). A servo-hydraulic test system MTS Landmark 370.10 with a contact extensometer model 634.31F-25 was used with a crosshead speed of 0.5 mm/min. The specimens were lightly grinded on the longitudinal surface at the reduced section, with silicon carbide sandpaper (grade 400) to apply the speckled pattern for the DIC method.

The speckled pattern was made with a matte black acrylic spray enamel, which was applied over the area of interest (reduced section). Different application distances and angles were tested to find the best combination and to obtain a speckled pattern with a good adherence and heterogeneous distribution. The best conditions for the spray application were an application distance of 300 mm from the specimen and an application angle of 45 degrees. The covered area of the speckled pattern in the ROI was quantified by converting the images in 8-bit gray scale and then analyzed by contrast between black and white zones (image analysis) using ImageJ 1.54i 03^®^ software. The analyzed area was approximately 140 mm^2^, whereas the speckled pattern area (black speckles) was roughly 52 mm^2^, given a ratio of 37% (Figure 3).

The images were acquired by using a Samsung Galaxy A52 ® cell phone, armed with a 64 megapixels camera set to record 30 frames per second (fps). This device was fixed on a tripod and turned on using a Bluetooth controller to avoid blur. The tests were recorded in HD quality video and uploaded into GOM Correlate^®^ 2022 free software for frames fragmentation. It was not necessary to use special lighting conditions for the measurement system, the diaphragm opening of the cell phone camera was enough to obtain the adequate contrast between black and clear areas of the speckle pattern. Figure 4 shows the experimental arrangement.

GOM Correlate^®^ has the possibility of defining a variety of virtual extensometers based on the reference distances on the full ROI in any direction and size. This feature was used to analyze the DIC reliability in the quasi-static and cyclic loading tests. In the case of the quasi-static loading, a virtual extensometer with the same length and which was close to the conventional physical extensometer was set into the analysis (see Figure 5).

Conventional tensile test data were processed into the Equations (2) and (3) to obtain the true stress ($\tilde{\sigma }$) as a function of the engineering strain (ε) behavior. This calculation was performed until the ultimate tensile strength $\sigma_{u}$ (before necking).
(2)$$\tilde{\sigma }=\sigma \left(1+\varepsilon \right)$$
(3)$$\tilde{\varepsilon }=\ln\left(1+\varepsilon \right)$$

The registered force by the tensile test, in conjunction with the final cross-sectional area of the specimen, was used to determine the strength at fracture (point of fracture). The strain measurements obtained from the virtual extensometer (DIC calculation) were compared with the true strain determined by Equation (3). A Hollomon constitutive model (Equation (4)) was used to determine the strength coefficient H, as well as the strain-hardening exponent n.
(4)$$\tilde{\sigma }=H\;{\left({\tilde{\varepsilon }}_{P}\right)}^{n}$$

To assess the DIC technique’s reliability during the cyclic loading tests, an increasing loading–unloading sequence was repeated five times on two specimens with the same geometry, as shown in Figure 2. A constant crosshead speed of 0.5 mm/min was used. In each of the five cycles, the specimens were tensile stretched at different stress levels above the yield strength and then immediately released to zero stress with the same crosshead speed. The remanent plastic strain ($\varepsilon_{p}$) was recorded and, subsequently, the specimen was again loaded in tension at a higher stress. Young’s modulus and yield strength ($\sigma_{0.2})$ were used to determine the elastic strain component ($\varepsilon_{e}$). 

Axial strain-controlled fatigue tests were conducted to analyze the contour maps of the full-field strain resolved by the DIC customized system and to correlate with the fatigue crack formation. The fatigue specimens (Figure 6), in accordance with the ISO 12106 standard [33], were machined in the LT to rolling direction. The specimens were ground using silicon carbide sandpapers (180–1200 grade) over the thickness at the reduce section. The servo-hydraulic test system MTS Landmark 370.10 was used again, but it was matched with an MTS extensometer model 632.13F-20 with a gauge length of 10 mm to control the strain amplitude during the fatigue tests.

The DIC-customized system was also used for the fatigue tests, but without the possibility to create a virtual extensometer of the same gauge length, so that the physical one with the rubber bands was used to attach it. Two strain amplitudes ($\varepsilon_{a}=0.006$ and $\varepsilon_{a}=0.008$) were defined for the fatigue test at a strain ratio $R_{E}=-1$, with a triangular waveform. The test frequency (f) was determined by the Equation (5), at a constant strain rate of $\dot{\varepsilon }=0.016\;s^{-1}$ for both strain amplitude conditions, with a video-recording rate of 30 fps, thus fulfilling the Nyquist relation ($f_{N}\;\geq \;2$) [34,35]. The failure criterion for the fatigue test was established at a 15% drop in the force from the stable strain hysteresis loop.
(5)$$f=\frac{\dot{\varepsilon }}{4\varepsilon_{a}}$$

## 3. Results and Discussion

### 3.1. Microstructure of the AL-6XN Material

Figure 7 shows the microstructure of the AL-6XN super-austenitic stainless steel along the longitudinal (L), short transverse (ST), and long transverse (LT) directions. 

The observed microstructure was homogeneous in the three analyzed sections, constituted by equiaxial and randomly oriented grains of the austenite γ phase (FCC) with an average grain size of approximately 20 µm. The presence of annealed twinning and deformation bands along the rolling direction were observed. The average hardnesses for the three sections are shown in Figure 7. As it was possible to observe, the L section direction slightly increased its hardness in comparison with ST and LT section directions. This aspect can be attributed to the presence of the oriented deformation bands.

### 3.2. Tensile Mechanical Properties

Figure 8 shows the conventional stress–strain behavior for the AL-6XN stainless steel (transverse to rolling direction). From Figure 8, the average tensile mechanical properties were determined (Table 3).

From the conventional stress–strain behavior (average curve), the true stress–strain curve was plotted (Figure 9). This curve was compared with those obtained by DIC, i.e., where the strain was measured by the virtual extensometer. As it is possible to observe in Figure 9a, the stress–strain curves determined by DIC fit very well with that obtained from the conventional measurements. However, a slight deviation appears when the deformation was no longer uniform, due to the necking formation at $\varepsilon_{u}\approx 0.38,$ according to the conventional stress–strain. In addition, an absolute DIC strain error was determined with respect to the measured strain by the extensometer $\varepsilon_{ext}$ as presented in Equation (6). Figure 9b presents this error as a function of the true stress. The absolute DIC strain error was close to zero over a true stress range up to approximately 350 MPa. This region represented the linear–elastic material behavior, where it was found that the DIC customized system and the extensometer provided similar results. For the plastic material behavior, an increment was observed for the absolute DIC strain error with some scatter. The worst error was lower than 3%, which rose at the necking zone formation.
(6)$$e=\left|\varepsilon_{DIC}-\varepsilon_{ext}\right|\times 100$$

### 3.3. Loading–Unloading Behavior

Figure 10 shows the loading–unloading curves for the five different tensile cycles applied to the AL-6XN material (the strain corresponds to the conventional physical extensometer). As it is possible to observe, the loading–unloading cycles follow the linear behavior stablished by the Young’s modulus, followed by a non-linear behavior because of the plastic deformation, whereas a hardening effect was presented from each tensile cycle to the next one, due to the accumulated residual plastic strain at the end of each cycle. The hardening effect followed the true stress–strain behavior according to the strength coefficient and strain-hardening exponent (Figure 9 and Table 3).

To observe the correspondence between the displacement measurements determined from the physical and virtual extensometer (DIC), the strain was plotted as a function of the time for each loading and unloading cycle (Figure 11a, Figure 12a, Figure 13a, Figure 14a and Figure 15a). Also, the images taken from the loading–unloading cycle tests are shown in Figure 11b, Figure 12b, Figure 13b, Figure 14b and Figure 15b. From these results, a very good approximation of the DIC measurements with the physical extensometer was observed. As can be noted, the residual strain ($\varepsilon_{r}$) increased due to the increment in the plastic strain component in each loading–unloading cycle. On the other hand, the mapping DIC images (Figure 11b, Figure 12b, Figure 13b, Figure 14b and Figure 15b) displayed a non-uniform displacement along the ROI. This could be associated with the speckles’ shape, which did not have a defined geometry, producing non-equal strain. DIC mapping represented a biaxial strain (*x*, *y*), whereas the extensometer measurements corresponds to the axial strain.

Once the strain results obtained by the DIC were verified for the quasi-static and loading–unloading test conditions against the physical extensometer, low cycle fatigue experiments (strain control) were carried out at two different strain amplitudes ($\varepsilon_{a}=0.008$ and $\varepsilon_{a}=0.006$). A DIC with full field strain measurements on the ROI was used. The objective of these experiments was to analyze the DIC’s reliability in detecting the crack location (fatigue damage) under a cyclic loading (tension–compression) imposed at a strain ratio $R_{\varepsilon }=-1$. According to the failure criterion (a 15% drop in force), the fatigue life at $\varepsilon_{a}=0.008$ and $\varepsilon_{a}=0.006$ was reached at 2502 ± 282.5 and 3494 ± 34.3 cycles, respectively. From these results, the DIC images for four different fatigue damage stages from both strain amplitudes were taken (Figure 16 and Figure 17), i.e., 25%, 50%, 75%, and final failure (15% drop in force).

The DIC strain mapping from the four fatigue damage levels (Figure 16 and Figure 17) was analyzed to observe the zones with a high strain concentration, which could eventually represent potential crack initiation sites. 

The specimen that fatigued at $\varepsilon_{a}=0.008$ showed a zone with a maximum strain value (determined from obeserved displacements) of about 0.010 (Figure 16a) at 25% of damage, which is higher than the nominal strain applied (0.008 ε). This zone tends to retain the highest strain values during the test, reaching a final value of roughly 0.015 at the last cycle (final failure), which is approximately two times higher than the nominal strain applied (Figure 16d). 

In the case of the specimen that fatigued at $\varepsilon_{a}=0.006$, from displacements of the DIC mapping, a maximum strain value of 0.014 (Figure 17a) at 25% of damage was reached, which is approximately two times the nominal strain applied. At the end of the fatigue life (100% damage), this zone reached a value of ε=0.026 at the last cycle, which is more than four times the nominal strain (Figure 17d). In this case, the fatigue crack was detected with the naked eye.

To observe the final condition of the fatigued specimens, the speckle pattern was removed (Figure 18 and Figure 19). The specimen tested at $\varepsilon_{a}=0.008$ (Figure 18) showed the fatigue crack, which seems to have started in the opposite surface of the speckle, growing through the thickness of the material.

Regarding the specimen tested at $\varepsilon_{a}=0.006$, the plastic strain with observed the naked eye was higher than the specimen tested at $\varepsilon_{a}=0.008$, which matched with the observed strain mapping during the fatigue tests. The fatigue crack also seems to have started in the opposite surface of the speckle and grew through the thickness until reaching the opposite surface (see Figure 19).

## 4. Conclusions

From the results reported in this work, the following statements can be drawn:DIC represents a good alternative for strain measurements during quasi-static and cyclic loadings. The true stress–strain curves generated by the virtual extensometer using the DIC technique provided accurate strain measurements, as verified against those measured by a conventional physical extensometer. A slight deviation appeared when the strain was no longer uniform due to the necking formation.The absolute DIC strain error follows a linear trend, with practically zero slopes over the linear elastic material behavior. However, once the plastic strain takes place, the absolute DIC strain error increases as a function of the true stress. The error was not larger than 3%.In the loading–unloading sequence test, the strain measurements provided by the virtual extensometer also adjusted very well with those provided by the physical extensometer. The DIC technique used was shown to be able to determine the residual strain in each subsequent cycle, which confirms its feasibility as an alternative measurement technique.For the strain control fatigue tests, the strain mapping allowed to determine zones with higher strain values than the nominal strain amplitude applied. These zones eventually could become potential crack initiation sites.The experimental set-up used demonstrates that DIC can be considered a low-cost technique for accurate strain measurements in the full ROI.

## Figures and Tables

**Figure 1 materials-17-03697-f001:**
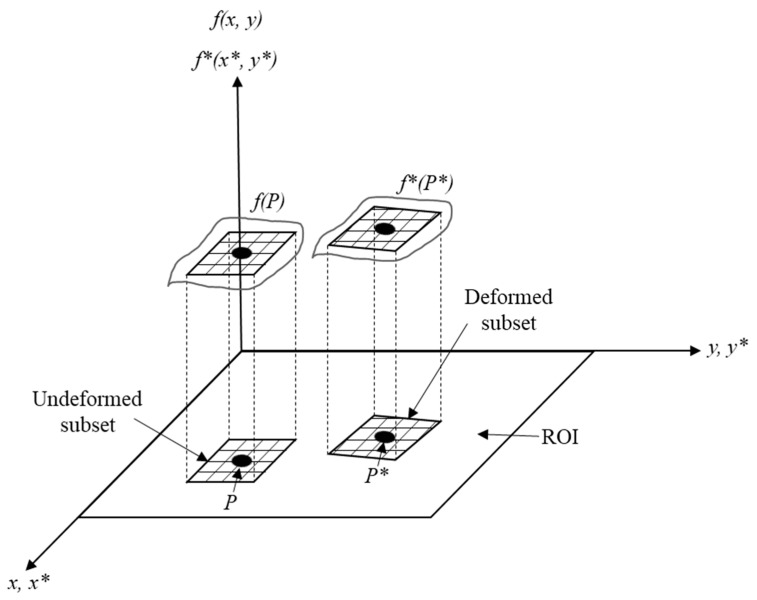
Schematic representation of the digital image correlation technique, adapted from [17,18,19].

**Figure 2 materials-17-03697-f002:**
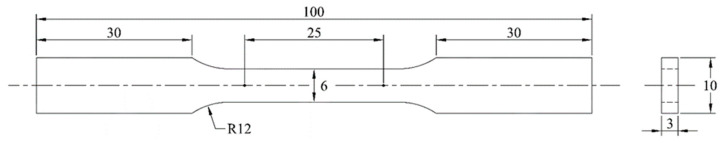
Scheme of specimen geometry for tensile test. Dimensions are in mm.

**Figure 3 materials-17-03697-f003:**
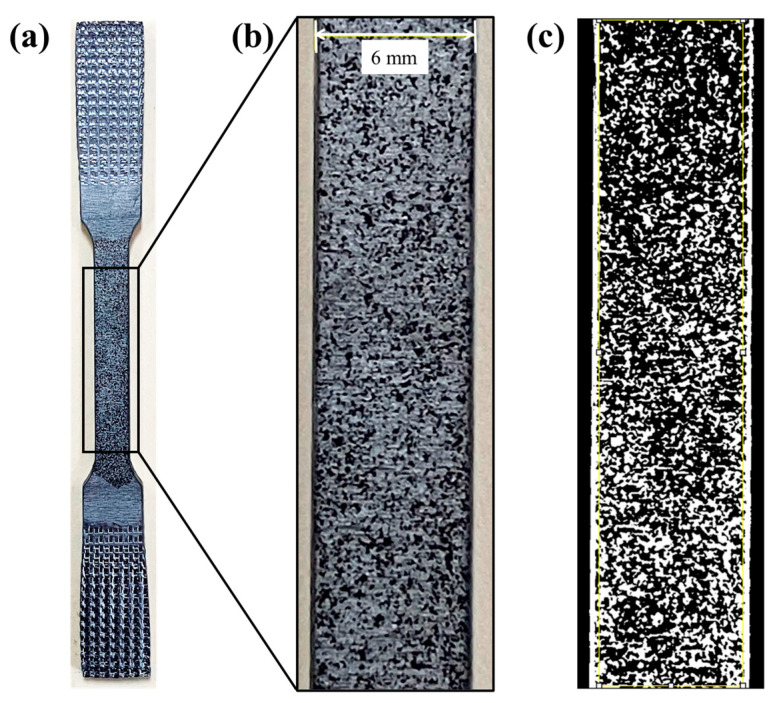
Speckle pattern used for digital image correlation analysis, (**a**) tensile specimen, (**b**) analyzed area, and (**c**) 8-bit image.

**Figure 4 materials-17-03697-f004:**
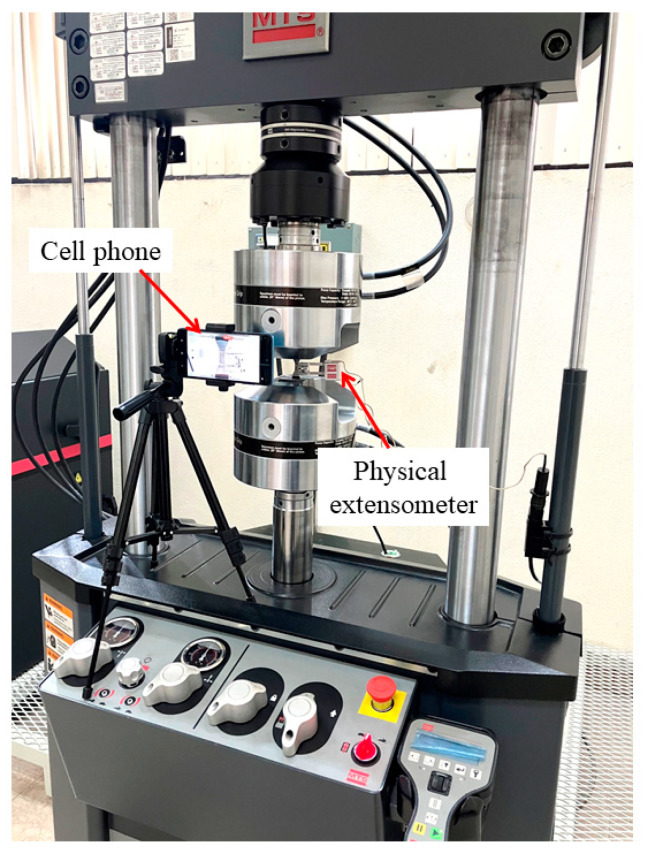
Experimental arrangement to use digital image correlation technique.

**Figure 5 materials-17-03697-f005:**
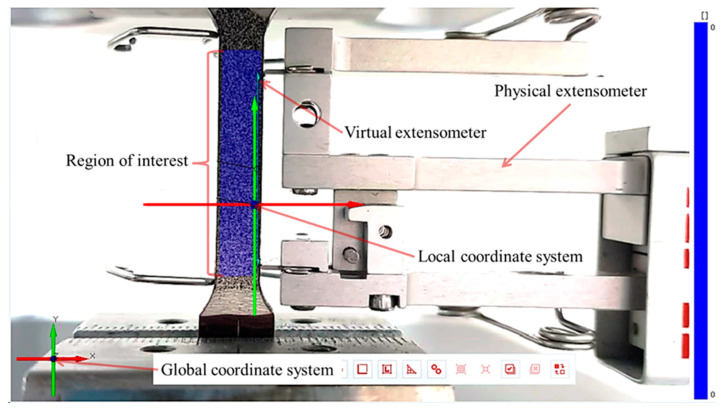
Virtual and physical extensometer for measurements comparison.

**Figure 6 materials-17-03697-f006:**
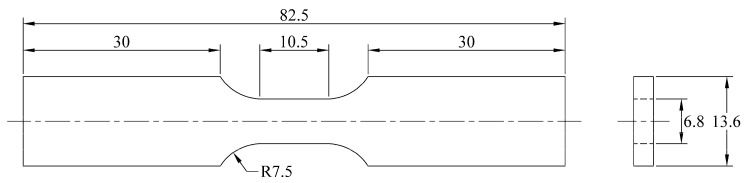
Specimen geometry for the strain-controlled fatigue tests. Dimensions are in mm.

**Figure 7 materials-17-03697-f007:**
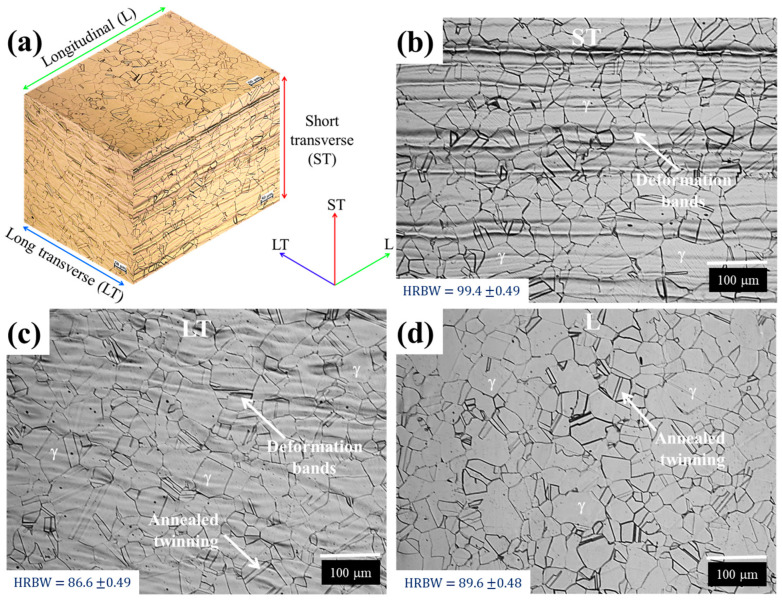
Microstructure of the AL-6XN stainless steel, (**a**) three dimensional overview, (**b**) longitudinal to rolling direction (L), (**c**) long transverse direction (LT), and (**d**) short transverse direction (ST).

**Figure 8 materials-17-03697-f008:**
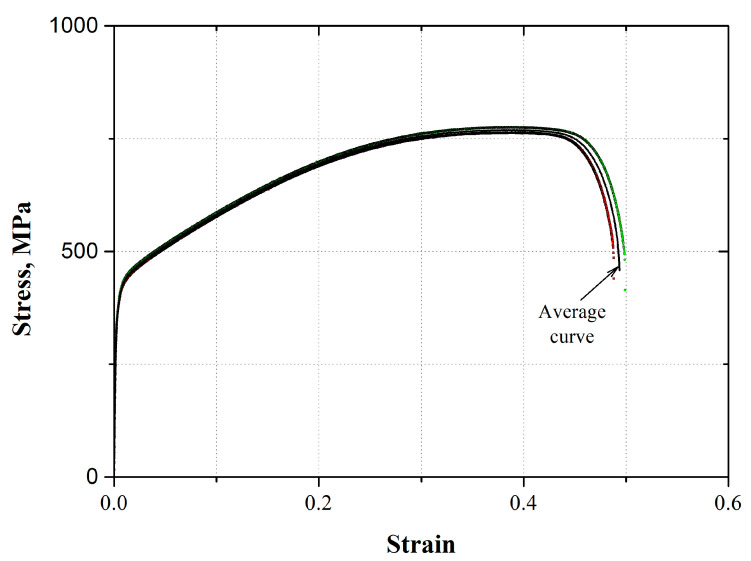
Conventional stress–strain behavior for the AL-6XN super austenitic stainless steel.

**Figure 9 materials-17-03697-f009:**
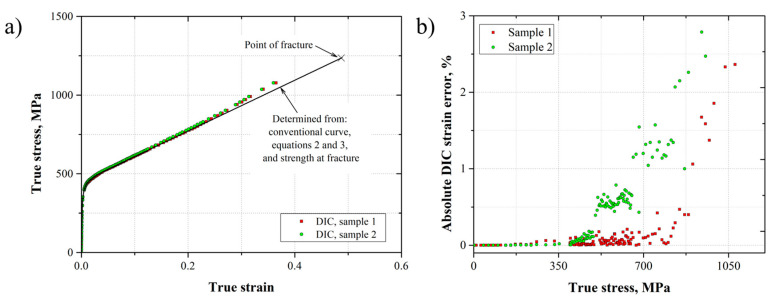
True stress–strain behavior of the AL-6XN stainless steel (**a**) and absolute DIC strain error (**b**).

**Figure 10 materials-17-03697-f010:**
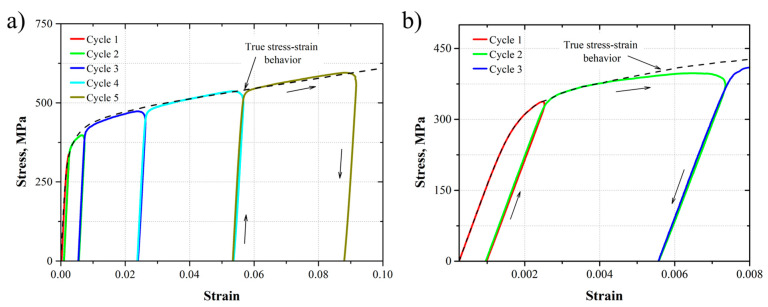
(**a**) Stress–strain behavior of the five loading–unloading cycles, (**b**) a zoom-in to the firsts loading–unloading cycles.

**Figure 11 materials-17-03697-f011:**
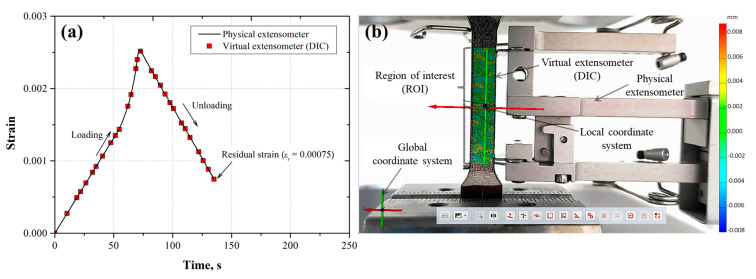
(**a**) First loading–unloading cycle ($\varepsilon_{max}=0.0025,\;\varepsilon_{r}=7.5\times 10^{-4}$) for the AL-6XN material, (**b**) mapping displacements at the end of the cycle.

**Figure 12 materials-17-03697-f012:**
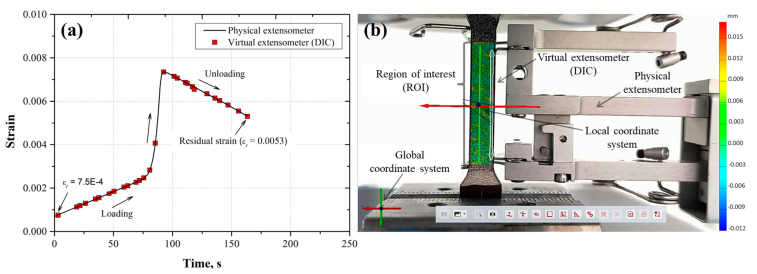
(**a**) Second loading–unloading cycle ($\varepsilon_{max}=0.0073,\;\varepsilon_{r}=5.3\times 10^{-3}$) for the AL-6XN material, (**b**) mapping displacements at the end of the cycle.

**Figure 13 materials-17-03697-f013:**
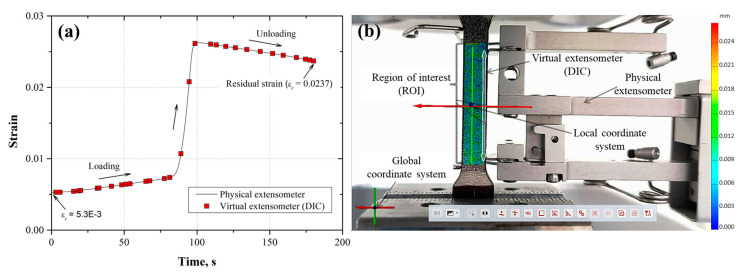
(**a**) Third loading–unloading cycle ($\varepsilon_{max}=0.0262,\;\varepsilon_{r}=2.37\times 10^{-2}$) for the AL-6XN material, (**b**) mapping displacements at the end of the cycle.

**Figure 14 materials-17-03697-f014:**
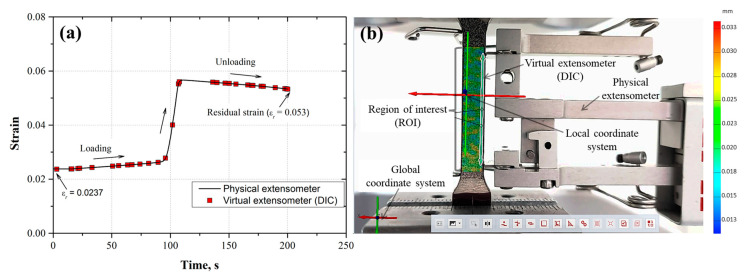
(**a**) Fourth loading–unloading cycle ($\varepsilon_{max}=0.0567,\;\varepsilon_{r}=5.3\times 10^{-2}$) for the AL-6XN material, (**b**) mapping displacements at the end of the cycle.

**Figure 15 materials-17-03697-f015:**
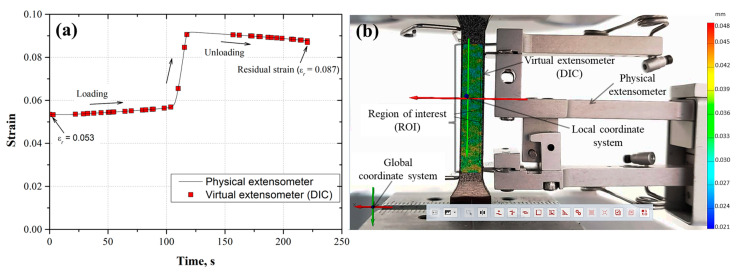
(**a**) Fifth loading–unloading cycle ($\varepsilon_{max}=0.0920,\;\varepsilon_{r}=8.7\times 10^{-2}$) for the AL-6XN material, (**b**) mapping displacements at the end of the cycle.

**Figure 16 materials-17-03697-f016:**
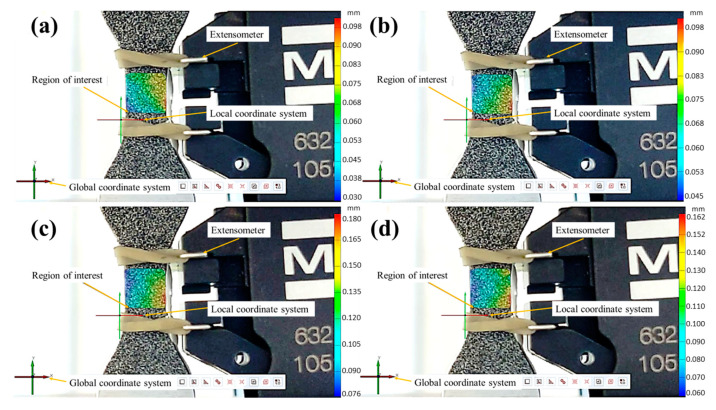
Fatigue specimen at $\varepsilon_{a}=0.008$, showing different fatigue life damage: (**a**) 25%, (**b**) 50%, (**c**) 75% FL, and (**d**) 100% (final failure).

**Figure 17 materials-17-03697-f017:**
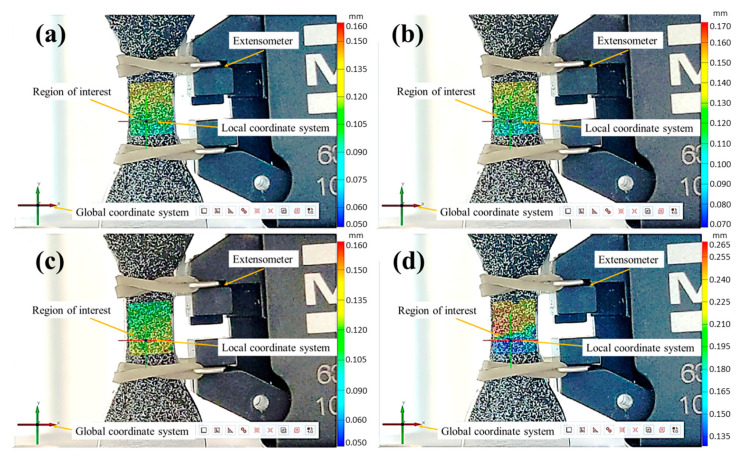
Fatigue specimen at $\varepsilon_{a}=0.006$, showing different fatigue life damage: (**a**) 25%, (**b**) 50%, (**c**) 75% FL, and (**d**) 100% (final failure).

**Figure 18 materials-17-03697-f018:**
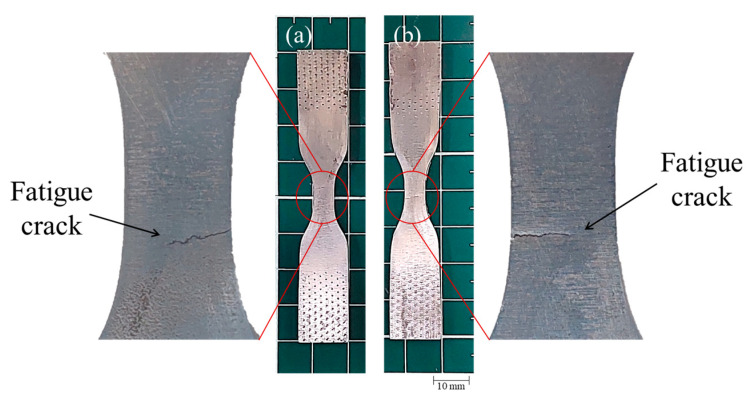
The crack observed in the fatigue specimen tested at $\varepsilon_{a}=0.008$, (**a**) speckled surface, (**b**) opposite surface of the speckle.

**Figure 19 materials-17-03697-f019:**
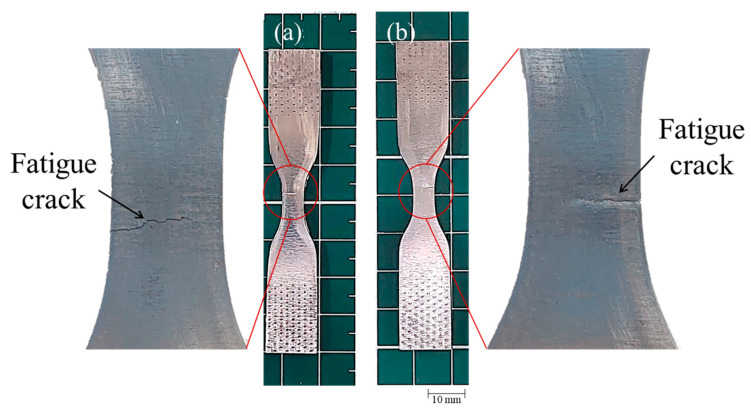
The crack observed in the fatigue specimen tested at $\varepsilon_{a}=0.006$, (**a**) speckled surface, (**b**) opposite surface of the speckle.

**Table 1 materials-17-03697-t001:** Overview of several methods for the monitoring of fatigue process in materials [16].

Method	Stage	Remarks
Potential drop method (PDM)	Fatigue crack	The calibration curves are geometry-dependent and, they must be developed for each case.
Acoustic emission (AE)	Fatigue crack	There are some undesired AE sources; rubbing between fracture surfaces and moving parts, hammering and vibrating.
Ultrasonic waves	Fatigue crack Fatigue damage	Parameters such as wave attenuation and sound velocity can be used to characterize the microstructural fatigue damage, which exhibits small changes and often with large plateaus.
Electric resistance	Fatigue damage	It can only be applicable for conductive materials. It requires several electrodes for properly map the fatigue damage accumulation.
Hardness measurements	Fatigue damage	Its application might be questionable because indentations can serve as notches. Polishing the material surface for microhardness removes the strain-hardened/softened surfaces.
X-ray diffraction	Fatigue damage	Its application as an in situ tool could represent a challenge. An initial dislocation structure will influence the parameters used for the fatigue damage analysis.
Thermometric measurements	Fatigue damage	Measurements are strongly dependent upon stress, frequency, and environmental conditions.
Strain-based	Fatigue damage	Loads must be applied to evaluate the produced strain and has limitation in practical applications.
Positron annihilation	Fatigue damage	It is a material-dependent method, and, in some cases, it could not be applicable to fatigue damage detection due to initial positron trapping sites.
Magnetic methods	Fatigue damage	It can only be applicable for ferromagnetic materials. It must be measurable without loading.

**Table 2 materials-17-03697-t002:** Chemical composition of AL-6XN stainless steel (weight%).

C	Mn	P	S	Si	Cr	Ni	Cu	Mo	N	Fe
0.017	0.490	0.030	0.0002	0.493	21.080	25.100	0.420	6.150	0.220	Bal

**Table 3 materials-17-03697-t003:** Tensile mechanical properties for the AL-6XN stainless steel.

	$\sigma_{0.2}$ (MPa)	$\sigma_{u}$ (MPa)	E (GPa)	$\varepsilon_{u}$	$\varepsilon_{f}$	H (MPa)	n	$U_{T}$ (MJm^−3^)
AL-6XN	357.6 ±30.8	769.5 ±8.3	183.7 ±5.1	0.38 ±0.005	0.5 ±0.01	1644.1 ±7.9	0.39 ±0.004	331.0 ± 24.0

$\sigma_{0.2}$ = yield strength at 0.2% strain, $\sigma_{u}$ = ultimate tensile strength (UTS), E = Young’s modulus, $\varepsilon_{u}=$ strain at UTS, $\varepsilon_{f}$ = strain at fracture, H = strength coefficient, n = strain hardening exponent, $U_{T}$ = toughness.

## Data Availability

The raw data supporting the conclusions of this article will be made available by the authors on request.

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
