# Peer review of "Strain Measurement during Quasi-Static and Cyclic Loads in AL-6XN Material Using Digital Image Correlation Technique"

_materials, 2024, doi:10.3390/ma17153697_

Round 1

Reviewer 1 Report

Comments and Suggestions for Authors

In experimental mechanics, the method of digital image correlation has become widespread for assessing the deformation of solids. The method is based on determining displacements through the procedure of minimizing the correlation coefficient as a measure of the similarity of image areas of the object’s surface before and after deformation. Currently, the digital image correlation method is one of the most common approaches to studying the processes of deformation and destruction of structurally inhomogeneous

materials. The authors of the paper implemented a customized digital image correlation system to monitor the deformation occurring in a 3.0 mm thick AL-6XN stainless steel cold-rolled plate subjected to quasi-static and cyclic load tests.

The system allows you to assess material damage in local areas, which is useful for science and practice, but there are several questions:

1. It is necessary to show a study of the metrological characteristics of the measurement system, as well as the calculation of its main parameters.

2. Does the developed method allow us to estimate displacements in images and evaluate them under conditions of significant changes in surface topography and significant deformation, for example 50%?

3. What should be the lighting conditions for the analysed surface?

4. Is the active formation of deformation relief on the measurement surface an obstacle to the correct assessment of deformations? Does the method calculate 2D or 3D deformations?

5. On what basis is the size of the correlation area selected?

6. Can the proposed method be used to assess the stages of deformation at the macro, meso and micro levels?

7. I recommend analysing the article in the Introduction:

https://www.tandfonline.com/doi/abs/10.1080/15421406.2017.1360700

Author Response

Strain measurement during quasi-static and cyclic loads in AL‑6XN material using digital image correlation technique

Donovan Ramírez-Acevedo 1, R.R. Ambriz 1,*, Ch.J. García 2, C. Mendoza 2 and D. Jaramillo 1

1 Instituto Politécnico Nacional CIITEC-IPN, Cerrada de Cecati S/N Col. Sta. Catarina, Azcapotzalco, Ciudad de México, México. C.P. 02250.

Corresponding author: [email protected]

Paper Submitted to Journal Materials

ID: 3091026

We would like to thank the editor for considering our work and giving us the opportunity to reply to the reviewer’s comments and submit a revised manuscript. We would also like to thank the reviewers for the comments, enabling us to enhance the content and the text of the manuscript. According to the reviewers remarks, modifications are applied to the revised manuscript. The changes in the manuscript are highlighted in yellow.

Responses to reviewers’ comments

Reviewer #1:

In experimental mechanics, the method of digital image correlation has become widespread for assessing the deformation of solids. The method is based on determining displacements through the procedure of minimizing the correlation coefficient as a measure of the similarity of image areas of the object’s surface before and after deformation. Currently, the digital image correlation method is one of the most common approaches to studying the processes of deformation and destruction of structurally inhomogeneous materials. The authors of the paper implemented a customized digital image correlation system to monitor the deformation occurring in a 3.0 mm thick AL-6XN stainless steel cold-rolled plate subjected to quasi-static and cyclic load tests.

The system allows you to assess material damage in local areas, which is useful for science and practice, but there are several questions:

It is necessary to show a study of the metrological characteristics of the measurement system, as well as the calculation of its main parameters.

R. The absolute DIC strain error was included. Please see Figure 9b.

Does the developed method allow us to estimate displacements in images and evaluate them under conditions of significant changes in surface topography and significant deformation, for example 50%?

R. The surface roughness for the AL-6XN material had not changed significantly during the tests. In this case, our used measurement system (DIC) is able to estimate displacement measurements corresponding with approximately 40% strain (see Figure 9). No different surface topographies were tested.

What should be the lighting conditions for the analysed surface?

R. It was not necessary to use special lighting conditions for the measurement system. The diaphragm opening of the cell phone camera was enough to obtain an adequate contrast between black and clear areas of the speckle pattern.

Is the active formation of deformation relief on the measurement surface an obstacle to the correct assessment of deformations? Does the method calculate 2D or 3D deformations?

R. As can be seen in Figures 11-15, the deformation relief is not an obstacle to determine the displacements measurements. It is to say that the used DIC technique showed to be able to determine the residual strain for each subsequent loading cycle. The developed method only applies to calculate 2D displacements. It is possible to determine 3D measurements by using GOM Correlate ® Free software, but the use of an additional camera is necessary.

On what basis is the size of the correlation area selected?

R. The selected area for the image correlation was selected according to reduced section of the samples, where the failure is expected.

Can the proposed method be used to assess the stages of deformation at the macro, meso and micro levels?

R. The proposed method was able to resolve displacements measurements in micrometers.

I recommend analysing the article in the Introduction:

https://www.tandfonline.com/doi/abs/10.1080/15421406.2017.1360700

R. Thank you for the recommendation, the reference has been considered.

Reviewer 2 Report

Comments and Suggestions for Authors

1.   In Figures 11 through 15, inconsistencies are noted in the size and positioning of the images. Specifically, the alignment of panels (a) and (b) lacks uniformity, which impacts visual coherence and aesthetic presentation.

2.   The text provides a comprehensive description of the experimental phenomena and results. However, it lacks an in-depth analysis of the conclusions derived from the experiments, and the ensuing discussion is deemed insufficiently detailed and informative.

3.   Additional determinants or conclusions should be incorporated to underscore the advantages of DIC (Digital Image Correlation) technology in order to enhance persuasiveness and facilitate a clearer thematic comprehension for the reader.

Comments on the Quality of English Language

1.   In Figures 11 through 15, inconsistencies are noted in the size and positioning of the images. Specifically, the alignment of panels (a) and (b) lacks uniformity, which impacts visual coherence and aesthetic presentation.

2.   The text provides a comprehensive description of the experimental phenomena and results. However, it lacks an in-depth analysis of the conclusions derived from the experiments, and the ensuing discussion is deemed insufficiently detailed and informative.

3.   Additional determinants or conclusions should be incorporated to underscore the advantages of DIC (Digital Image Correlation) technology in order to enhance persuasiveness and facilitate a clearer thematic comprehension for the reader.

Author Response

Strain measurement during quasi-static and cyclic loads in AL‑6XN material using digital image correlation technique

Donovan Ramírez-Acevedo 1, R.R. Ambriz 1,*, Ch.J. García 2, C. Mendoza 2 and D. Jaramillo 1

1 Instituto Politécnico Nacional CIITEC-IPN, Cerrada de Cecati S/N Col. Sta. Catarina, Azcapotzalco, Ciudad de México, México. C.P. 02250.

Corresponding author: [email protected]

Paper Submitted to Journal Materials

ID: 3091026

We would like to thank the editor for considering our work and giving us the opportunity to reply to the reviewer’s comments and submit a revised manuscript. We would also like to thank the reviewers for the comments, enabling us to enhance the content and the text of the manuscript. According to the reviewers remarks, modifications are applied to the revised manuscript. The changes in the manuscript are highlighted in yellow.

Reviewer #2:

In Figures 11 through 15, inconsistencies are noted in the size and positioning of the images. Specifically, the alignment of panels (a) and (b) lacks uniformity, which impacts visual coherence and aesthetic presentation.

R. Sorry for this inconvenience, the inconsistencies were amended.

The text provides a comprehensive description of the experimental phenomena and results. However, it lacks an in-depth analysis of the conclusions derived from the experiments, and the ensuing discussion is deemed insufficiently detailed and informative.

R. To support the conclusions a metrological characteristic was added (Figure 9b). The absolute DIC error was determined, which is lower than 3%.

Additional determinants or conclusions should be incorporated to underscore the advantages of DIC (Digital Image Correlation) technology in order to enhance persuasiveness and facilitate a clearer thematic comprehension for the reader.

R. Thank you for your recommendation. However, the authors believed that conclusions provided the key points for DIC, such as a good alternative for strain measurements during quasi-static and cyclic loadings, for strain fatigue damage monitoring and crack initiation detection.

Reviewer 3 Report

Comments and Suggestions for Authors

The article may be interesting to the researchers, who use DIC method. However, the novelty of the article is low. In my opinion, the only added value of the article is that DIC was applied in a combination with a really low-cost data acquisition equipment (i.e. the smart phone). Beside that, some additional issues need to be addressed:

1.) Lines 75-84: the references for an application of a DIC method for fatigue loads are rather old. I sincerely doubt that nothing was published in this field in the last 10 years. Please, see reference of Litrop et al. (doi: 10.1016/j.engfailanal.2022.106495) for application of DIC to measure fatigue crack propagation in low-cycle fatigue tests.

2.) Line 177: "The DIC customized system..." => What system are the authors talking about? Which freeware software?

3.) Figure 9: So, which of the two diagrams is more realistic/correct? Please, give an opinion with a reasonable explanation.

4.) Figures 11-15: How were this experiments carried out? With a load control? Where in the text can this information be found? Namely, such a strain-time diagrams cannot be obtained using strain- or displacement-controlled tests.

5.) Lines 291 and 309: "... at necked eye" => "... with necked eye"

6.) Figure 18: If I understand it right, DIC was not used for crack detection, only for strain measurement? Why not, if it works well for both tasks?

Comments on the Quality of English Language

Minor editing of English language required.

Author Response

Strain measurement during quasi-static and cyclic loads in AL‑6XN material using digital image correlation technique

Donovan Ramírez-Acevedo 1, R.R. Ambriz 1,*, Ch.J. García 2, C. Mendoza 2 and D. Jaramillo 1

1 Instituto Politécnico Nacional CIITEC-IPN, Cerrada de Cecati S/N Col. Sta. Catarina, Azcapotzalco, Ciudad de México, México. C.P. 02250.

Corresponding author: [email protected]

Paper Submitted to Journal Materials

ID: 3091026

We would like to thank the editor for considering our work and giving us the opportunity to reply to the reviewer’s comments and submit a revised manuscript. We would also like to thank the reviewers for the comments, enabling us to enhance the content and the text of the manuscript. According to the reviewers remarks, modifications are applied to the revised manuscript. The changes in the manuscript are highlighted in yellow.

Responses to reviewers’ comments

Reviewer #3:

The article may be interesting to the researchers, who use DIC method. However, the novelty of the article is low. In my opinion, the only added value of the article is that DIC was applied in a combination with a really low-cost data acquisition equipment (i.e. the smart phone). Beside that, some additional issues need to be addressed:

Lines 75-84: the references for an application of a DIC method for fatigue loads are rather old. I sincerely doubt that nothing was published in this field in the last 10 years. Please, see reference of Litrop et al. (doi: 10.1016/j.engfailanal.2022.106495) for application of DIC to measure fatigue crack propagation in low-cycle fatigue tests.

R. The references used were analyzed according with the objective of the work. We appreciate the provided reference, which was incorporated.

Line 177: "The DIC customized system..." => What system are the authors talking about? Which freeware software?

R. The customized system was comprised by a cellphone camera with video recorder, speckle pattern, and the use of GOM Correlate® free software version. This system was described in the second section of the article.

Figure 9: So, which of the two diagrams is more realistic/correct? Please, give an opinion with a reasonable explanation.

R. Both diagrams are representative of the tensile mechanical behavior of the AL-6XN material. As a reference, typical coefficients of variation can be found in the literature (Mechanical behavior of materials. Norman E. Dowling).

Figures 11-15: How were this experiments carried out? With a load control? Where in the text can this information be found? Namely, such a strain-time diagrams cannot be obtained using strain- or displacement-controlled tests.

R. A constant crosshead speed of 0.5 mm/min was used for these experiments until reach the stress level. The time for each cycle was determined by synchronizing the video record with the test.

Lines 291 and 309: "... at necked eye" => "... with necked eye"

R. Thank you for your comment. This inconvenience was corrected.

Figure 18: If I understand it right, DIC was not used for crack detection, only for strain measurement? Why not, if it works well for both tasks?

R. DIC was used for strain measurement for tensile and loading-unloading tests. Afterwards, DIC was used to identify potential crack initiation points based on the strain mapping. This is described in the article.

Round 2

Reviewer 1 Report

Comments and Suggestions for Authors

Accept.